# A Novel PMDI Fiber Optic Hydrophone Incorporating IOC-Based Phase Modulator

**Chunxi Zhang [1], Sufan Yang [1,2,*] and Xiaxiao Wang [1]**

1   School of Instrumentation and Optoelectronic Engineering, Beihang University, Beijing 100191, China; zhangchunxi@buaa.edu.cn (C.Z.); wangxiaxiao@buaa.edu.cn (X.W.)
2   Shenyuan Honors College, Beihang University, Beijing 100191, China
*   Correspondence: yangsufan@buaa.edu.cn

**Abstract:** Fiber-optic hydrophone (FOH) has significant potential in many applications of hydroacoustic sensing and underwater communication. A novel path-matched differential interferometer fiber optic hydrophone (PMDI-FOH) approach incorporating an integrated-optic component (IOC) is presented in this paper. It is presented to meet the demands for high-quality dynamic measurements, which solves the problems with the conventional homodyne detection system's low modulation frequency. The IOC functions as a phase-generated carrier (PGC) component. The scheme is investigated both in theory and experiments. The theoretical and experimental results verify the effectiveness of the proposed scheme. It achieves a high SNR of up to 20.29 dB demodulations. The proposed system is cost-effective and has excellent potential in building next-generation underwater sensing and communication networks.

**Keywords:** path-matched differential interferometer fiber optic hydrophones (PMDI-FOH); integrated-optic component (IOC); phase-generated carrier (PGC); homodyne detection





## 1. Introduction

Fiber-optic hydrophone (FOH), a novel underwater acoustic signal sensor based on fiber optic and optoelectronic technology, has been researched extensively. It converts hydroacoustic signals into optical signals through optical detection approaches and transmits them to the signal processor through the optical fiber to extract acoustic signal information. It has significant potential in hydroacoustic sensing [1–3] and underwater communication applications [4,5]. Due to its superior features, including immunity to electromagnetic interference, high sensitivity, wide dynamic range, and small size, it is an effective substitute for the traditional piezoelectric ceramic sensor [6,7].

FOHs can be divided into polarization type, intensity (amplitude), frequency, interference, and fiber-grating type by different sensing schemes [3,8–10]. The interferometric FOH is the most promising and studied [10]. The basic principle of interferometric FOHs is that the interferometer effectively converts the external sound field information into phase change information [11].

With a proper phase detection method, phase change information can be extracted. The widely accepted phase detection techniques are based on providing the sensing information encoded on a carrier signal, and include the phase generated carrier (PGC) demodulation method [12–16], heterodyne demodulation method [17–19], and 3 × 3 diversity detection method [20,21]. PGC is widely adopted due to its wide dynamic range, high sensitivity, and good linearity [22]. A homodyne detection method includes a broad class of interrogation approaches [8]. Usually, a sinusoidally modulated optical carrier generates a signal in an interferometric FOH system. A high modulation frequency is required when the signal to be measured is high-frequency and has a large amplitude [22]. The commonly used PGC component for an interferometric FOH is the piezoelectric ceramic (PZT) [9,23]. However, the modulation frequency of the PZT is usually as low as dozens of kHz. Moreover, the

PZT has several disadvantages, such as being bulky, expensive, and having low-level integration. Since the modulation frequency is limited by the modulation bandwidth of the modulation device, it is possible to use a modulation device with a large modulation bandwidth. Several works on phase modulators have recently been developed to meet the demands of PGC-based FOH systems, such as the thin piezoelectric sheet (TPS) [24]. The TPS has some drawbacks. Although the TPS is highly integrated and miniature, its modulation bandwidth is limited to 50–300 Hz [24]. What is more, it needs a high voltage to drive. A Lithium Niobate (LiNbO3) component has a low drive voltage and an ultra-high modulation bandwidth of up to hundreds of GHz [25–27], providing an alternative research idea.

In addition, multi-sensor operation is critical to using FOHs in most practical applications [9]. The multi-sensor structures can be divided into two categories: (a) separate and (b) multiplexed architectures [28]. Based on the different interferometer architectures, the separate architectures of FOHs can be further divided into four main configurations: Mach–Zehnder interferometer (MZI), Michelson interferometer (MI), Fabry–Perot interferometer (FPI), and Sagnac interferometer (SI), among which MZI and MI are widely used due to their simple structure, high sensitivity, and easy array formation [8,10,29]. To form a multiplexing system, time-division multiplexing (TDM), frequency-division multiplexing (FDM), wavelength-division multiplexing (WDM), code-division multiplexing (CDM), and space-division multiplexing (SDM) are the most commonly used multiplexing technologies [10]. Path-matched interferometer fiber optic hydrophones (PMDI-FOH), as s kind of simplified TDM architecture [9], are commonly used in the FOH multiplexing system [30,31]. It has become an attractive alternative to conventional Michelson-FOH due to its many features [9,32], including a simple "wet-end" structure, high light energy utilization, and low phase noise [30,32].

A novel PGC-based PMDI-FOH design incorporating an integrated-optic component (IOC) is presented in this paper. The IOC device is a semi-custom phase modulator fabricated in Y-cut Z-propagation LiNbO3 by Titanium-indiffused technology. It is a critical component in the scheme, used as a PGC component. Compared with PZT, the IOC modulator achieves a much higher modulation frequency. Theoretical analysis and experiments are conducted to verify the effectiveness of the scheme. The proposed system achieves a high SNR of up to 20.29 dB demodulations. It solves the problems with a conventional homodyne detection system's low modulation frequency. The excellent experimental results suggest that the scheme has good potential in large dynamic and high-frequency detection applications.

## 2. System Design and Principle

### 2.1. PMDI-FOH System Design

To simplify the array architecture, a path-matched differential interferometer (PMDI). is used in multiplexing FOHs. The PMDI-FOH achieves equal-arm interference by time delay matching. It consists of three major components: (a) a compensating interferometer (CIF) which converts modulations into phase modulations; (b) a sensing array which converts the acoustic pressure into a measurable parameter in amplitude, wavelength, or frequency of the light passing through the fiber through respective modulations; and (c) a phase demodulator which interrogates the phase modulations encountered by the interferometer.

The proposed PMDI-FOH system using a PMDI structure is shown in Figure 1. The light emerging from a narrow-linewidth laser (NLL) passes through an optical isolator. Then, it enters an acoustic-optic modulator (AOM) to generate an optical pulse. The AOM is switched by a programmable pulse generator (PPG) with a duty cycle, which is determined by the number of sensors in the sensing array. Passing through the AOM, the pulse is split by a 50:50 PM coupler into an unbalanced Mach–Zehnder interferometer (MZI). The MZI, as the CIF, generates periodic dual pulses. The short path propagates through the IOC phase modulator (IOC MOD), and the long path is through a delay coil. The length

of the coil is 2*L*. The IOC MOD is modulated by an arbitrary function generator (AFG). The generated dual pulse injects into the sensing array through an optical circulator. The sensing array consists of several in-line Michelson interferometers (MIs). It uses several Faraday rotation mirrors (FRMs), $1 \times 2$ couplers, and delay coils. The length of each delay coil is L. They are also used as sensors in the system, labeled $S_1$ ... $S_{n-2}$, $S_{n-1}$, $S_n$, and $S_{n+1}$. The couplers are labeled $C_1$ ... $C_{n-2}$, $C_{n-1}$, $C_n$, and $C_{n+1}$. Additionally, the FRMs are labeled $FRM_1$ ... $FRM_{n-2}$, $FRM_{n-1}$, $FRM_n$, and $FRM_{n+1}$. Notably, the FRMs applying a double pass efficiently cancel all polarization rotation effects in the delay fibers [9].

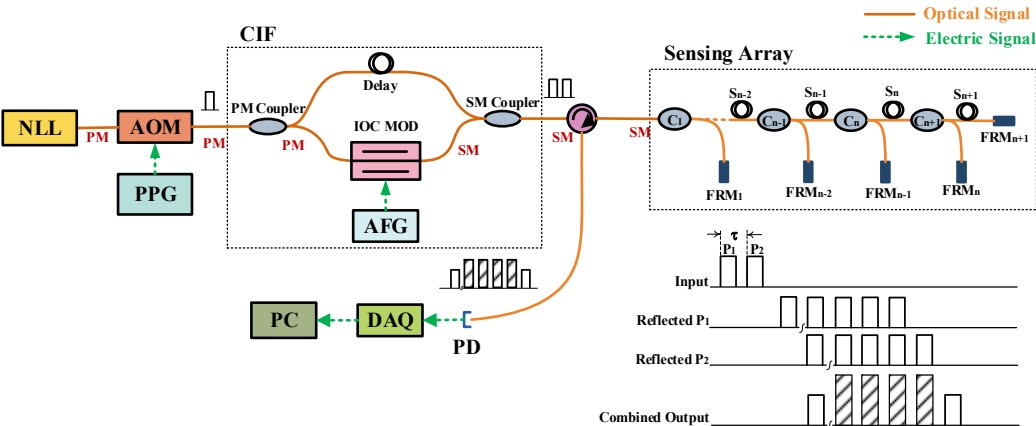

**Figure 1.** Schematic of the IOC-based PMDI-FOH system (PM: polarization-maintaining, SM: single mode.

A pulse time sequence is formed after the sensing array returns a series of pulses. The interval of each pulse is $\tau = \frac{2nL}{c}$. In each pulse trace, the reflected pulse $P_1$ and reflected pulse $P_2$ overlap in the time sequence. For the reflected pulse $P_1$ an $P_2$, the optical field can be expressed as follows [33]:

$$E_1 = A_1 e^{j[\omega t + \varphi_s(t) + \varphi_1]} \tag{1}$$

$$E_2 = A_2 e^{j[\omega t + \varphi_m(t) + \varphi_2]} \tag{2}$$

where $A_1$ and $A_2$ are the amplitude of optical waves, $\omega$ is the frequency for the laser source, $\varphi_1$ and $\varphi_2$ are the initial phases of $E_1$ and $E_2$, respectively, $\varphi_s(t)$ is the time-dependent phase signal of interest, and $\varphi_m(t)$ is the phase change introduced by the IOC modulator. The two overlapped pulses interfere [33]:

$$I = \langle (E_1 + E_2)(E_1 + E_2)^* \rangle = A_1^2 + A_2^2 + 2A_1 A_2 \cos[\varphi_s(t) - \varphi_m(t) + \varphi_0] \tag{3}$$

where $\varphi_0 = \varphi_1 - \varphi_2$ is the initial phase for the interference pulse.

Thus, the interference pulses carry sensing and phase-modulated information in the photodetector (PD). Acquired by the high-speed data acquisition (DAQ) card, the data are transferred to a personal computer (PC) and processed. With a proper phase detection method, the acoustic signals can be demodulated.

Additionally, it is essential to consider how pulse timing affects signal interference in a PMDI-FOH system. Figure 2 depicts the section of the sensor array in Figure 1. An unbalanced MI is created between positions B and C as a separate sensor, enclosing the $S_n$ with $FRM_n$ and $FRM_{n1}$, respectively. Due to the double pass time of the latter through each sensor, the first pulse $P_1$ is ahead of the second pulse $P_2$ by the separation time $\tau_s = \frac{\tau}{2}$. At time t, pulse $P_1$ enters the interferometer at position B, and pulse $P_2$ enters the subnetwork at position A. The pulses will span at any time. Thus, the two pulse-span sensors, $S_{n-1}$ and $S_{n-2}$, will span. Then, the time differential interferometer is formed between A and C. It is clear that pulse $P_2$ carries no phase information with respect to A at time t, while pulse $P_1$ acquires a phase $\varphi_{S_{n-1}}(t) + \varphi_{S_{n-2}}(t)$ at B. At time $t + \tau_s$, the

first pulse P$_1$, initially at B, has traveled twice through sensor S$_n$ after reflecting from the FRM$_n$ to merge at position D with the second pulse P2. The pulse P$_2$ has traveled through S$_{n-2}$ and S$_{n-1}$ and has reflected in mirror FRM$_{n-1}$. At time $t + \tau_s$, the phase received by P$_1$ through the interferometer is $2\varphi_{S_{n-1}}(t + \tau_s)$, while the phase received by P$_2$ is $\varphi_{S_{n-1}}(t + \tau_s) + \varphi_{S_{n-2}}(t + \tau_s)$. In the detector, the common-mode phase information received by the two pulses returning together to the PD at location D is differenced, leaving only the net phase collected across the interferometer arms.

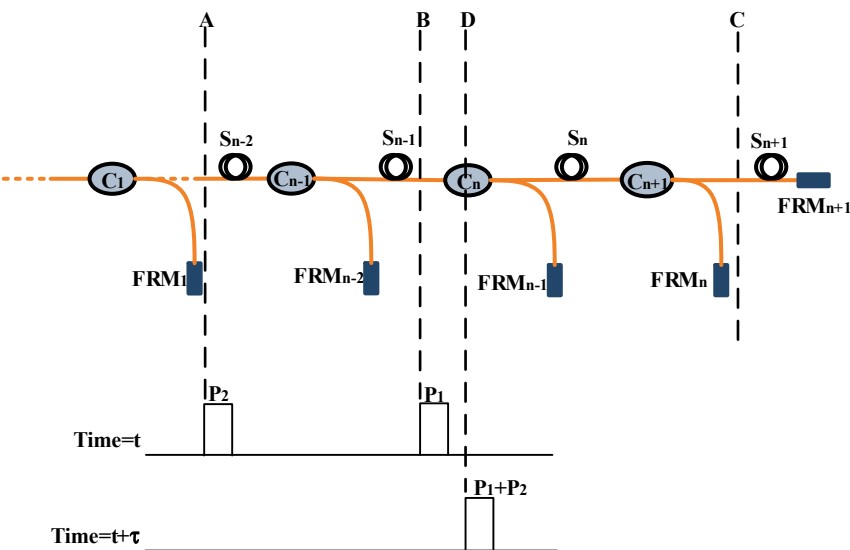

**Figure 2.** Section of a sensing array.

However, the optical switches, such as the AOM that generate the light pulse, have a limited extinction ratio (ER), and background or leakage light is present after the device is turned "off". The leakage light enters in the sensor array, which introduces cross-talk noises in the PMDI-FOH system. For a TDM system, the finite ER is the main cause of cross-talk noise [9,34]. A high "on-off" ER optical switch of 50 dB or more is required for large-scale and high-performance FOH array systems [9,35]. The proposed system shown in Figure 1 uses the AOM as the optical switch. Thus, the ER of the AOM is required to be higher than 50 dB.

### 2.2. IOC-Based Phase Modulator

In the PDMI-FOH system, the IOC-based phase modulator is the critical component. It is used as a PGC component. It differs from conventional PGC-based systems, usually PZT [23,24]. It is a semi-custom device fabricated in Y-cut Z-propagation LiNbO3 by Titanium-indiffused technology. It consists of a single, through the optical waveguide. Although Y-cut crystals have lower sensitivity to an electric field than Z-cut ones, they are proven to achieve suitable temperature stability [36–38].

The mechanism of the phase modulator is the Pockels effect. A voltage $V$ is applied to the waveguide's electrodes, which causes a change in the refractive index of the LiNbO3 crystals. When the optical waves pass through the waveguide, the phase changes are proportional to the amplitude of the voltage. After the light polarized along the Z-axis passes through the optical waveguide, the phase change caused by the electro-optical effect is [30,39]:

$$\varphi_m = \pi n_e^3 \gamma_{22} \Gamma \frac{Vl}{G\lambda} \tag{4}$$

where $n_e$ is the extraordinary refractive index, $\gamma_{22}$ is the electro-optic coefficient of the LiNbO3 crystals, and $\Gamma$ is an electro-optic overlap integral factor. It implies the strength of the interaction between the optical and electric fields. Usually, it is between 0 and 1. $l$ is the

length of the electrodes, $G$ is electrode gap width, and $\lambda$ is the wavelength of the optical light in the waveguide.

If the definition of the half-wave voltage of the LiNbO3 waveguide is [40]:

$$V_\pi = \frac{\lambda G}{n_e^3 \gamma_{22} \Gamma} \tag{5}$$

Then, Equation (4) can be simplified as [30,40]:

$$\varphi_m = \pi \frac{V}{V_\pi} \tag{6}$$

### 2.3. PGC-Based Demodulation Method

The PGC demodulation technique extracts the phase information from the proposed PMDI-FOH system. The PGC technique introduces phase modulation outside the signal bandwidth. The modulation signal carries the signal of interest to sidebands and upconverts them. Therefore, the low-frequency noise is eliminated [12].

Suppose a sinusoidal modulation signal applied to the IOC modulator can be expressed as [9,23]:

$$V = V_{DC} + V_m \cos(\omega_m t) \tag{7}$$

where $V_{DC}$ is the DC bias voltage of the signal and $V_m$ and $\omega_m$ are the peak amplitude and modulation frequency of the sinusoidal wave, separately. Substituting Equation (7) into Equation (6), the resulting equation can be written as:

$$\varphi_m = \pi \frac{V_{DC}}{V_\pi} + \pi \frac{V_m}{V_\pi} \cos(\omega_m t) \tag{8}$$

Then, the output of the interferometer from the same time slot in the pulse train (corresponding to a particular sensor) is [9]:

$$\begin{aligned} I &= A + B\cos\left[\pi \tfrac{V_m}{V_\pi}\cos(\omega_m t) + \varphi_s(t) + \varphi_{0m}\right] \\ &= A + B\cos[C\cos(\omega_m t) + \varphi_s(t) + \varphi_{0m}] \end{aligned} \tag{9}$$

where A is a constant term proportional to the optical power, B is a coefficient related to optical power and interferometer visibility, $C = \pi \frac{V_m}{V_\pi}$ is a constant, which is called modulation depth, $\varphi_0$ is the initial phase mentioned above, and $\varphi_{0m} = \varphi_0 + \pi \frac{V_{DC}}{V_\pi}$ is the initial phase noise with a phase modulation.

The cosine term is written in terms of the Bessel expression [41]:

$$\begin{aligned} &\cos[C\cos(\omega_m t) + \varphi_s(t) + \varphi_{0m}] \\ &= \left[J_0(C) + 2\sum_{k=1}^{\infty}(-1)^k J_{2k}(C)\cos(2k\omega_m t)\right]\cos(\varphi_s(t) + \varphi_{0m}) \\ &\quad - \left[2\sum_{k=1}^{\infty}(-1)^{k-1} J_{2k-1}(C)\cos(2k-1)\omega_m t\right]\sin(\varphi_s(t) + \varphi_{0m}) \end{aligned} \tag{10}$$

where $J_n(C)$ is the first kind of n-order Bessel function.

As is shown in Figure 3, the digitized PD signal is split into two streams in the PGC-Arctangent technique. They are multiplied by $\cos(\omega_m t)$ and $\cos(2\omega_m t)$ separately, which is called up conversion [42]. After that, the mixing results are low-pass filtered to obtain in-phase signal $I_I$ and quadrature one $I_Q$, which be given as [10]:

$$I_I = -BJ_1(C)\sin(\varphi_s(t) + \varphi_{0m}) \tag{11}$$

$$I_Q = -BJ_2(C)\cos(\varphi_s(t) + \varphi_{0m}) \tag{12}$$

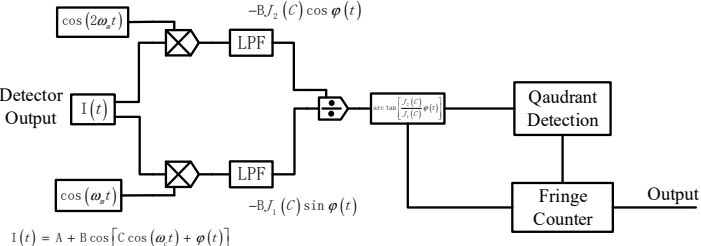

**Figure 3.** Schematic of the PGC-Arctangent demodulation technique [14,43].

The phase information can be recovered from the arctangent of the ratio of $I_Q$ to $I_I$, then [10]:

$$\varphi_s(t) + \varphi_0 = \arctan\left(\frac{I_I}{I_Q}\right) = \arctan\left(\frac{J_1(C)\sin(\varphi_s(t) + \varphi_{0\mathrm{m}})}{J_2(C)\cos(\varphi_s(t) + \varphi_{0\mathrm{m}})}\right) \tag{13}$$

As $\varphi_{0\mathrm{m}}$ is nonlinearly changed with the phase difference between the arms of the interferometer, and the cosine term represents the nonlinear relation of phase change to the interferometer output [37], it drifts slowly with time. After a properly high-pass filter, $\varphi_s(t)$ can be retrieved.

The extracted phase $\varphi_s(t)$ is proportional to $\frac{J_1(C)}{J_2(C)}$. It can be inferred that the PGC-Arctangent methods can avoid experiencing light intensity fluctuations in the light intensity, as they can maintain the modulation depth. If the modulation depth $C$ is set to 2.63, then $\frac{J_1(C)}{J_2(C)} = 1$. That is to say, the amplitude from the first and second harmonic components of the cosine term are equal, as is shown in Figure 4. As a result, the amplitude of the in-phase and quadrature terms are equal.

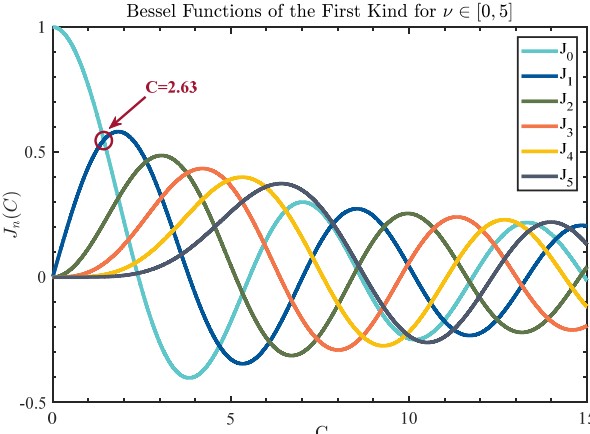

**Figure 4.** Six values of n-order first kind of Bessel function [9].

A crucial restriction of the arctangent method, when the quadrature sine and cosine components are transformed to phase, is that the signal phase can only change by a maximum of $\pi$ radians between samples. As shown in Figure 5, when the phase exceeds $\pi$ or is less than $-\pi$, an uncertainty is presented about whether the flag traveled clockwise or counter-clockwise around the circle. As a rule for demodulators, a shorter way was taken [3]. Thus, the demodulated phase abruptly jumps by $2\pi$. This is known as phase wrapping. Phase wrapping causes a disturbance in the continuity of the demodulated signal. For small amplitude phase signals, this does not affect the signal waveform. However, for significant amplitude phase signals, this adds additional noise to the detected phase signal. As a result, the dynamic range of FOH is limited to one unit circle period, i.e., $2\pi$.

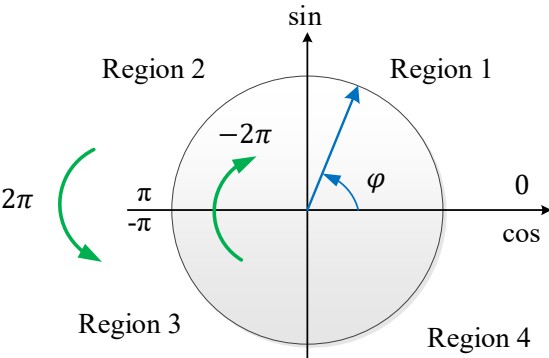

**Figure 5.** Operational diagram of fringe counting for correcting significant phase signals [14,23].

To correct the phase wrapping error, the fringe counter is used in the algorithm [22,43]. As shown in Figure 5 and Table 1 [43], the unit circle is divided into four regions according to the signs of in-phase and quadrature terms. The discrete baseband sequence $\sin \varphi([k])$, $\cos \varphi([k])$, $\sin \varphi([k-1])$, and $\cos \varphi([k-1])$ are stored in memory. Once a transition from Region 2 to Region 3 occurs, the fringe counter value increases by $2\pi$. From Region 3 to Region 2, the fringe count value is decreased by $2\pi$. The fringe counter value accumulates in the arctangent operation process. Theoretically, the dynamic range is increased as the fringe counter value increases.

**Table 1.** Look-up table for arctangent operation [23].

| Region | $\sin \varphi([k])$ | $\cos \varphi([k])$ | $\varphi_s(t) \in (-\pi, \pi)$ |
|--------|---------------------|---------------------|-------------------------------|
| 1 | >0 | >0 | $\tan^{-1}\left(\frac{\sin \varphi([k])}{\cos \varphi([k])}\right) - \pi$ |
| 2 | >0 | <0 | $\tan^{-1}\left(\frac{\sin \varphi([k])}{\cos \varphi([k])}\right)$ |
| 3 | <0 | <0 | $\tan^{-1}\left(\frac{\sin \varphi([k])}{\cos \varphi([k])}\right)$ |
| 4 | <0 | >0 | $\tan^{-1}\left(\frac{\sin \varphi([k])}{\cos \varphi([k])}\right) + \pi$ |

## 3. Experimental Methods and Results

### 3.1. Phase-Generated Carrier Measurement Experiment

Based on the principal analysis in Section 2, an experimental set-up for phase-generated carrier signal measurements is built firstly to verify the effectiveness of the IOC-based phase modulation, as shown in Figure 6. A NLL with 17 mW output emitting at 1550.92 nm is used. The continuous light is generated from the NLL and it is passed through an unbalanced MZI. The unbalanced MZI consists of a PM coupler, a semi-custom IOC device, a SM delay coil, and a SM coupler. The length of the delay coil is 50 m. The measured data for the IOC used can be seen in Table 2. The AFG modulates the IOC modulator and a sinusoidally modulated signal is applied to the IOC device. To realize $C = 2.63$ mentioned in Section 2, the peak modulation amplitude is set to 3.14 V with different modulation frequencies. The interferometric signal is received at the PD. The bandwidth of the PD is 200 MHz. The interferometric signal is captured by a high-speed DAQ card and processed on a personal computer (PC). The sampling rate of the DAQ card is 50 MHz.

The phase-generated carrier measurement experimental results are shown in Figures 7 and 8. The time domain results of the interferometric signals are shown at the top of the figures. The interferometric signal's power spectral density (PSD) analysis results are displayed at the bottom. Compared with Figures 7 and 8, limited by the sampling frequency, the harmonic terms decrease as the phase modulation frequency increases. The modulation frequency should be at least two times the maximum signal frequency, and the interferometric signal spectrum contains baseband and harmonics frequencies. It can be concluded that the IOC-based system has the potential for accurate acoustic measurement

because the harmonics of the frequency are sufficiently tiny with high modulation frequency. As the sampling rate is required at least eight times the modulation frequency for the PGC method [44], the modulation frequency of the IOC modulator in acoustic sensing experiments is set to 5 MHz.

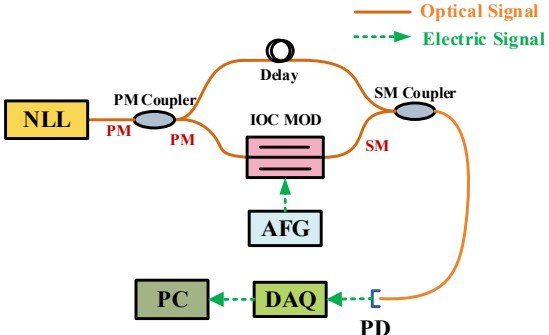

**Figure 6.** Schematic of experiment setup for phase-generated carrier signal measurement.

**Table 2.** Measurement data for IOC phase modulator.

| Parameters | Measured Data | Unit |
|---|---|---|
| average insertion loss | 3.44 | dB |
| polarization dependent loss | 0.54 | dB |
| output polarized crosstalk | −31.0 | dB |
| half-wave voltage (TE) | 3.75 | V |
| bandwidth | ≥300 | MHz |

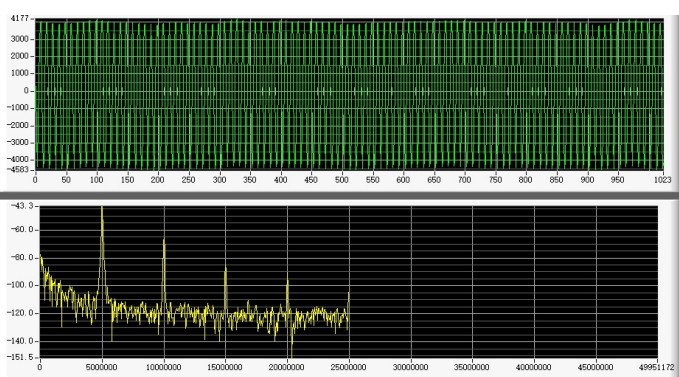

**Figure 7.** The phase-generated carrier measurement results when the modulated frequency is 5 MHz in the time and frequency domain.

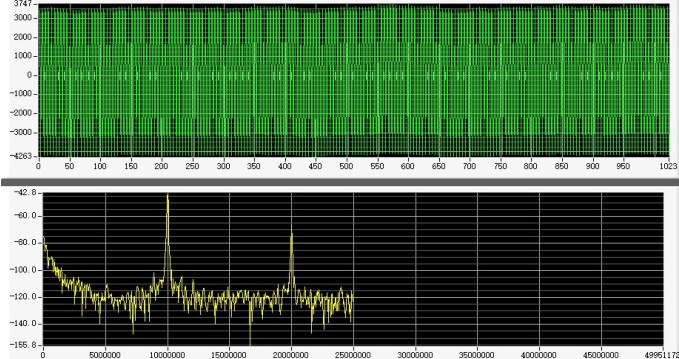

**Figure 8.** The phase-generated carrier measurement results when the modulated frequency is 10 MHz in the time and frequency domain.

### 3.2. Dual Pulses Measurement Experiments

Then, the dual pulse measurement experiment setup is established, as shown in Figure 9. A high extinction ratio (ER = 60.06 dB) AOM is used. The AOM converts the continuous light into a pulse. The AOM is modulated by a programmable pulse generator (PPG). The AFG sinusoidally modulates the IOC modulator. The PD receives the output waveform from the unbalanced MZI.

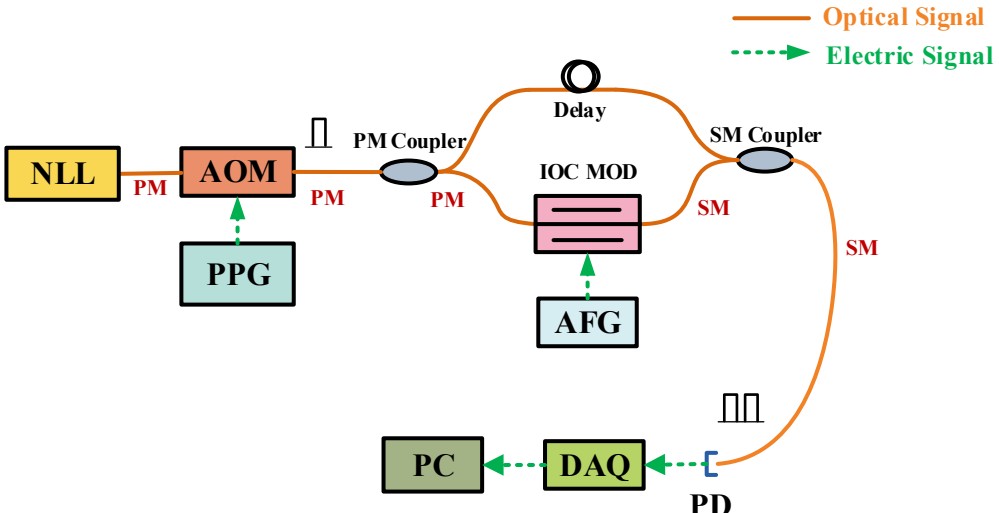

**Figure 9.** Schematic of experiment setup for dual pulses measurement.

The dual pulse measurement results are shown in Figure 10, when the pulse repetition frequency is 160 kHz with different pulse widths.

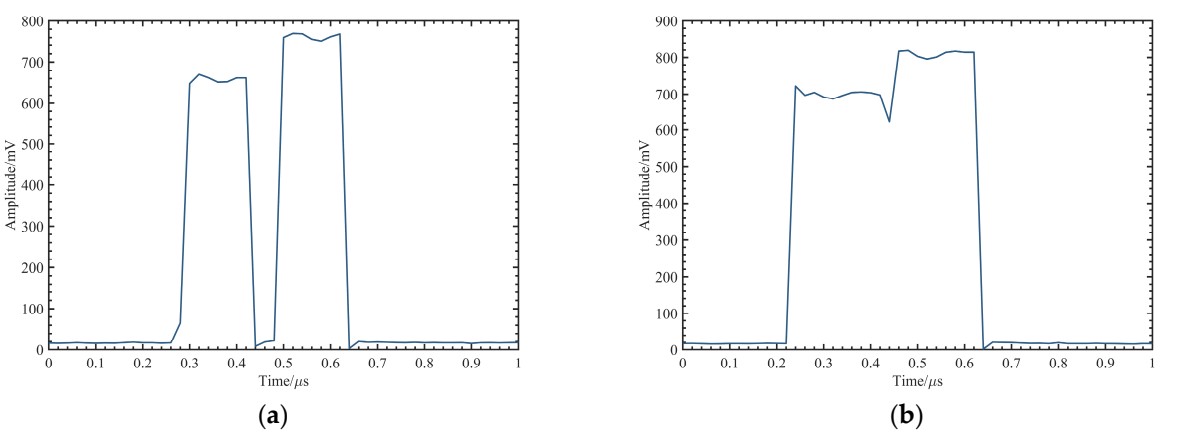

**Figure 10.** The output waveform of MZI generated dual pulses when the repetition frequency is 160 kHz, and the pulse width is (**a**) 150 ns and (**b**) 200 ns.

### 3.3. Acoustic Sensing Experiments

Acoustic sensing experiments are conducted with one sensing unit to verify the proposed scheme, as shown in Figure 11. The system consists of an unbalanced MZI and a sensing unit. The path difference of the MZI is 50 m and an attenuator is inserted in the long arm to introduce optical loss. The sensing unit comprises a SM coupler, a SM delay coil, a PZT phase stretcher, and two FRMs. A PZT phase stretcher is placed at the sensing arm of the MI. A continuous sinusoidal modulation signal is given to the PZT with different frequencies and amplitudes. Strain modulations are introduced to PZT, equivalent to an acoustic signal. The modulation index of the PZT is 1.9 rad/V < 5 kHz. The PD receives optical phase changes. The optical phase changes represent acoustic pressure

signals. Acquired by the DAQ card, the data are transferred to the computer and processed. The PGC-Arctangent method is used in the PC's demodulator, and the demodulated results are displayed on the PC.

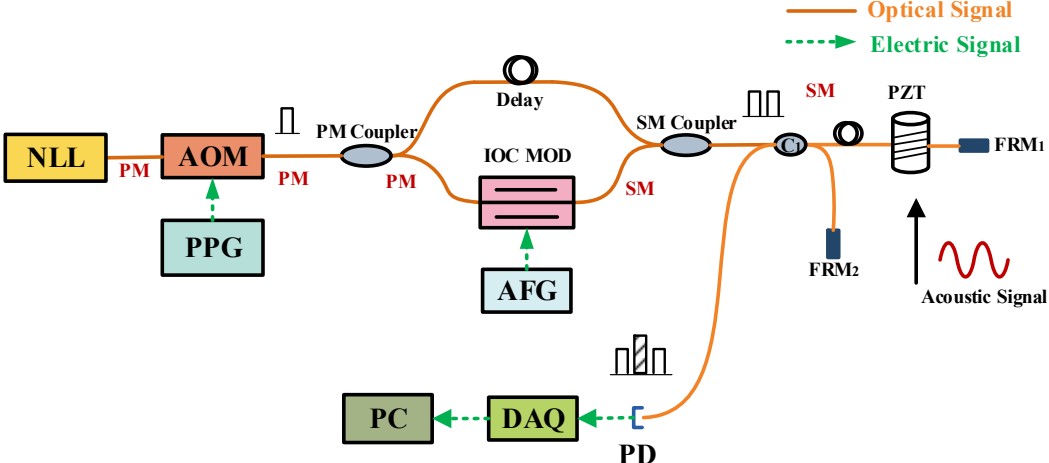

**Figure 11.** Schematic of experiment setup for acoustic signal measurement (PM: polarization-maintaining, SM: single mode).

When the pulse width is set to 200 ns, the interferometric pulses are received by PD, as shown in Figure 12. Each pair of interferometric pulses contains three pulses.

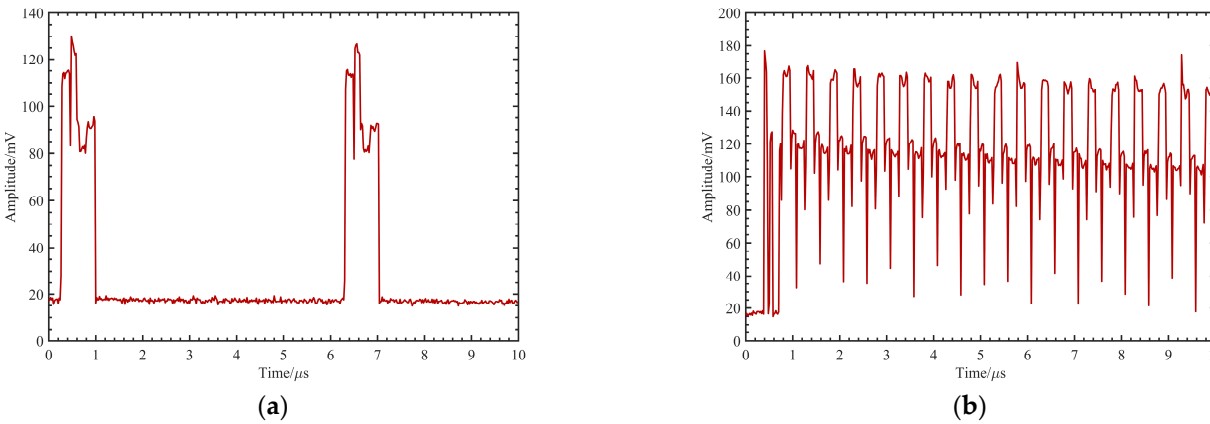

(**a**)　　　　　　　　　　　　　　　　　　　　　　　　　(**b**)

**Figure 12.** The output waveform of PMDI interferometric pulses when repetition frequency is (**a**) 160 kHz and (**b**) 1.4 MHz with 200 ns pulse width.

The acoustic sensing experiments are conducted as shown in Figure 11. The pulse width and repetition frequency are set to 200 ns and 160 kHz, separately. The IOC device is sinusoidally modulated with a peak amplitude of 3.14 V when the modulation frequency is 5 MHz. The demodulation results are shown in Figures 13–15, which demonstrate that a high SNR of 20.28 dB can be achieved. Figure 15 shows that signals can be demodulated at a frequency of 1 kHz with an amplitude of approximately 22.8 rad. Furthermore, the SNR can reach 18.48 dB. Moreover, it also can be seen from the demodulation results that the SNR slightly decreases when the signal intensity rises.

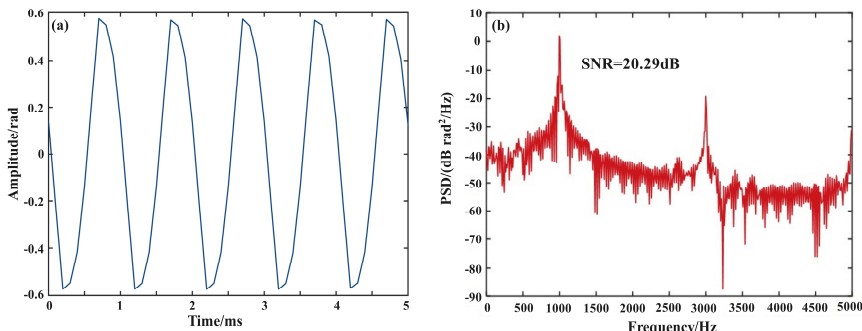

**Figure 13.** Output waveform and spectrum of PGC demodulation when the testing signal of 1 kHz with an amplitude of about 1.14 rad: (**a**) in time domain, (**b**) in frequency domain.

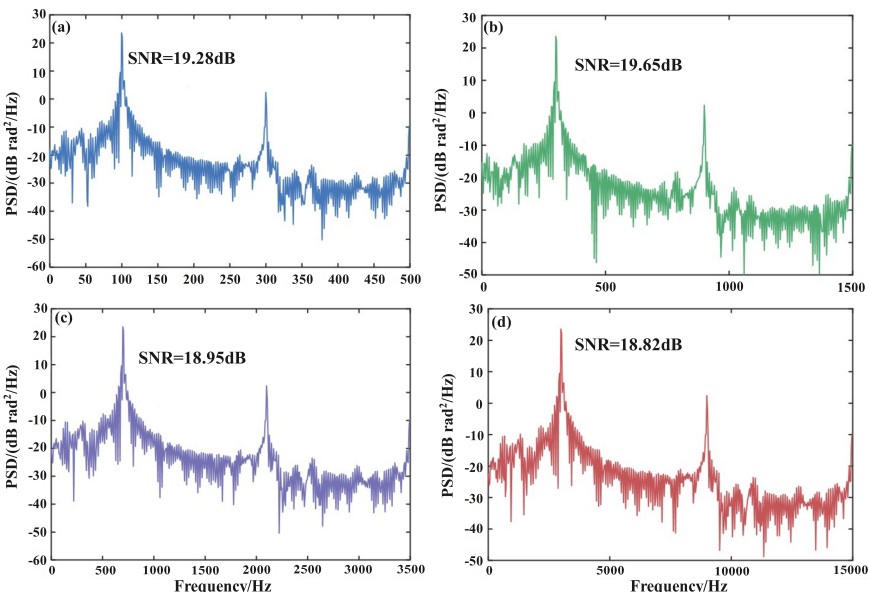

**Figure 14.** Output waveforms and spectrums of PGC demodulation with the amplitude of about 1.14 rad when the modulation frequency is (**a**) 100 Hz, (**b**) 300 Hz, (**c**) 700 Hz, and (**d**) 3 kHz, separately.

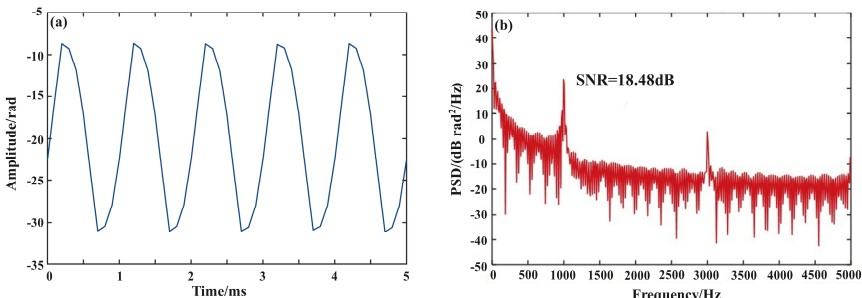

**Figure 15.** The output waveform and spectrum of PGC demodulation when the testing signal is 1 kHz with an amplitude of about 22.8 rad: (**a**) in time domain, (**b**) in frequency domain.

## 4. Discussion

A novel PMDI-FOH design using an IOC modulator is proposed in this paper. The critical parameter of the device is the modulation bandwidth. A conclusion is proposed by combining theory and simulation: the proposed scheme can primarily improve the system's detectable acoustic frequency and upper limits of the dynamic range due to its high modulation bandwidth. The system's detectable frequency and upper limit of dynamic range mainly depend on the carrier frequency and the demodulation technique [22]. The

maximum amplitude that can be demodulated has the following relationship with the modulation frequency [9]:

$$D_{max} = \frac{f_m}{2f_{sig}} \tag{14}$$

where $D_{max}$ is the maximum amplitude of the demodulated acoustic signal in radians, $D_{max}$ also represents the upper limit of dynamic range, $f_{sig}$ is the frequency of the acoustic signal, and $f_m$ is the modulation frequency.

The simulation effect of the modulation frequency on the dynamic range is shown in Figure 16. When the modulation frequency is increased, the detectable frequency and the upper limit of the dynamic range are increased. The IOC-based modulator can achieve a high modulation frequency due to its high bandwidth. Therefore, the proposed scheme demonstrates advantages in large dynamic range and high frequency in hydroacoustic detection applications.

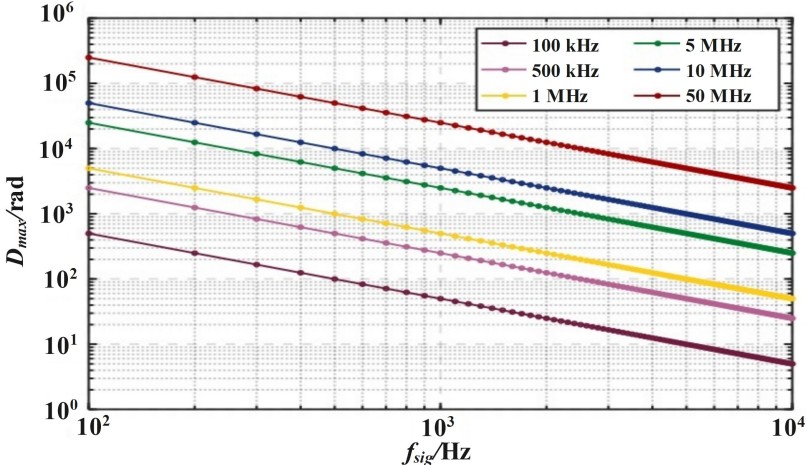

**Figure 16.** Maximum amplitude versus modulated frequency for the PGC method.

Recently, quantum communication has gained increasing popularity. Fiber optic hydrophones (FOHs) possess the potential to extract valuable modulation information in communication systems [5]. The proposed scheme with high modulation frequency shows potential in underwater quantum communication [45].

## 5. Conclusions

In this paper, a novel design of FOH incorporating an IOC-based phase modulator is developed. Experimental results verify the feasibility of the proposed scheme. The system achieves a high SNR of up to 20.29 dB demodulation results. Theoretically, increasing the modulation frequency of the phase modulator is beneficial to improve the dynamic range. Since the high modulation frequency and SNR are achievable, the proposed scheme can mutually facilitate building next-generation underwater sensing and communication networks.

**Author Contributions:** Conceptualization, C.Z. and S.Y.; methodology, S.Y.; validation, S.Y.; formal analysis, S.Y.; investigation, S.Y. and X.W.; data curation, S.Y.; writing—original draft preparation, S.Y.; writing—review and editing, C.Z. and X.W.; project administration. All authors have read and agreed to the published version of the manuscript.

**Funding:** This research received no external funding.

**Institutional Review Board Statement:** Not applicable.

**Informed Consent Statement:** Not applicable.

**Data Availability Statement:** The data presented in this study are available on reasonable request from the corresponding author.

**Conflicts of Interest:** The authors declare no conflict of interest.

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
