# Peer review of "A Novel PMDI Fiber Optic Hydrophone Incorporating IOC-Based Phase Modulator"

_photonics, doi:10.3390/photonics10080911_

Round 1

Reviewer 1 Report

The authors present a path matched differential interferometer configuration incorporating an integrated-optic component for fiber optic hydrophones. They claim their approach is feasible for realizing a multi-sensor FOH system, as shown in Figure 1 of the manuscript. They, however, provided a theoretical analysis and experimental results of only a single FOH sensor, as shown in Figure 10, which makes no significant contribution to the existing literature. 

The multi-sensor operation is actually the issue of most interest. Thus, additional theoretical analysis and experimental verification are needed, addressing the unambiguous extraction of individual sensor's signal and the expected sensor cross talk in the proposed FOH system.

Other than that, I have the following comments:

1. In figure 1, the time ?s is not defined, is it the same as the delay ? in line 96 or a different one? AFG is not defined in the text beforehand, it is defined afterwards on line 197.

2. The same symbol, L, is used for both the length of each delay coil and the length of the IOC modulator electrodes, included in equation (4). 

3. The first term in brackets on the first line in equation (8), ????/??, disappeared without introducing any equivalent term on the second line of the equation.

4. On line 246, the modulation index of PZT is not supposed to be compared to its modulation bandwidth.

5. On line 296, what is meant by "the system's single-frequency characteristic". Does it mean that the sensor linearity? 

Although the text is readable, many editing mistakes need to be corrected. For example, on lines: 52, 113, 181, 215, 246, 251, 256, 303 and 313.  

Author Response

We sincerely thank you for your careful and critical review of our manuscript. We are also grateful for your insightful suggestions, which allowed us to improve our manuscript significantly.

However, the value of our work does not lie in the multi-sensor operation. The real value of our work lies in using the IOC phase modulator in the path-matched differential fiber optic hydrophone (PMDI-FOH). And it covers several important intersecting topics, such as optical scheme organization, IOC-based devices, appropriate choice of phase modulation-related parameters, and even phase-detection methods. The manuscript verifies the effectiveness of using the IOC phase modulator in PDMI-FOH. We agree that multi-sensor operation is essential for practical applications. It is not the point of this manuscript. We will validate the effectiveness of our proposed scheme in the FOH array in our further work.

In light of these comments, we have conducted textual and graphical changes to the manuscript. Revisions and changes to the manuscript are highlighted in red. Our point-by-point response is as attached.

Reviewer 2 Report

“A Novel PMDI Fiber Optic Hydrophone Incorporating IOC-Based Phase Modulator” submitted for publication in Photonics presents a novel path-matched differential interferometer fiber optic hydrophone (PMDI-FOH) approach incorporating an integrated-optic component (IOC). The theoretical and experimental results verify the effectiveness of the proposed scheme. It achieves a high SNR of up to 20.29 dB demodulation. The proposed system is cost-effective and has excellent potential in large dynamic range and high-frequency hydroacoustic signal detection. However, there are some elements that are still unclear, and the manuscript lacks some significant research contents. The manuscript should be modified and improved. My suggestions are as follows (see list below).

1. In Figure 1, the PM is denoted as polarization-maintaining and phase modulation. It may be better use the full spelling to denote phase modulator.

2. The definition of phi_s(t) should be provided under Equation (1) where it first appears, and the definition of C1, C2 and C3 are missing. In line 94, the period is missing from the sentence.

3. There is an inconsistency between the length difference of the two arms of the interferometer in Figure 1 and the corresponding text. This discrepancy should be addressed and made consistent.

4. In Figure 15, the color used for the 50 kHz modulation frequency is similar to that of 50 MHz, making it difficult to differentiate between them. Furthermore, the variable represented by f_sig in the figure is not defined, and its meaning should be clarified. In line 102, the information regarding the sensing and phase modulation is not clear and needs to be further explained for better understanding.

5. Recently, quantum communication, which relies on the security provided by quantum mechanics, has been gaining increasing popularity. Fiber optic hydrophones (FOHs) possess the potential to extract valuable modulation information in communication systems. Therefore, the application of FOH and phase modulation technology developed in this manuscript in quantum communication holds significant importance and should not be overlooked. Therefore, I suggest that the authors consider citing the following articles [Rev. Mod. Phys. 94, 025008 (2022); National Science Review 10, nwac228 (2023)] which may make this manuscript be better.

6. Regarding lines 143-144, the concept of phase noise requires additional elaboration to enhance comprehension.

7. According to the authors, a higher modulation frequency enables the detection of higher-frequency signals of interest. In the acoustic sensing experiments, a modulation frequency of 5 MHz was chosen, while the testing signal was set at 1 kHz. To showcase the advantages of their work, it would be beneficial for the authors to utilize a higher-frequency testing signal rather than a 1 kHz signal.

Once the authors have addressed the points raised, I will agree that this work can be published in Photonics.

Moderate editing of English language required

Author Response

We sincerely thank you for your careful and critical review of our manuscript. We are also grateful for your encouraging opinions and insightful suggestions that allowed us to improve our manuscript substantially. In light of these comments, we have conducted many textual changes to the manuscript. Revisions and changes to the manuscript are highlighted in red. Our point-by-point response is as attached.

Reviewer 3 Report

The manuscript presents a fiber optic hydrophone scheme with an integrated-optic phase modulator. The scheme is realized experimentally with one sensing hydrophone, and its characteristics are analyzed. The paper is well written; however, there is a question of feasibility of this scheme with more than one sensing hydrophone. Figure 1 shows a scheme with 3 sensing hydrophones. The output signal is an interference from A and B, B and C, C and D. If there are only A and B, we have situation of Fig. 10, which has been studied. How can this scheme work if the output contains interference from B and C, C and D? What about power loss at couplers C2 and C3? If the authors could explain this, the manuscript could be accepted with revision. Additional comments: - optical isolator is not shown in Fig. 1; - SM coupled (Fig. 1) is not mentioned in the text; - Eq.1: what is phi_s(t)? Where it comes from? - Eq.3: what are angle brackets? - Eq.8: why pi*V_DC/V_pi disappeared? - which component of Fig. 1 contains the scheme of Fig. 2? - Eq. 12 is valid only when J1=J2; what if J1 is not equal J2? - Fig.2: should it be phi_s(t)? - p.8, line 246: dimensions rad/V=Hz. Is this correct? - there is no need for symbols on the lines in Fig. 15.

Author Response

We sincerely thank you for your careful and critical review of our manuscript. We are also grateful for your encouraging comments and insightful suggestions that allowed us to improve our manuscript substantially.

From your comments, it can be inferred that your review report was based on the original manuscript. Your review report is in Round 1. From now, other reviewers are in Round 2. And the manuscript has been revised once. Some changes have been made, and type errors have been fixed. This response letter will cover your concerns and mention the changes already made. Revisions and changes to the manuscript are highlighted in red. Following is our point-by-point response.

Round 2

Reviewer 1 Report

Addressing the unambiguous extraction of individual sensor's signal and the expected sensor cross talk in the proposed FOH system is important and is still needed.

The authors assume the DC component of the modulating voltage ??? = 0, which is not justified. 

The two added references [5] and [43] of the revised manuscript are the same.

No issues detected.

Author Response

Thank you for your suggestion. The point-to-point response is as attached.

Reviewer 2 Report

This manuscript can be accepted for publication. The authors must examine the references carefully during the proofreading stage.

  •  

 Minor editing of English language required

Author Response

Thank you for your encouraging words. We have sorry for the wrong citation of reference[5][Rev. Mod. Phys. 94, 025008 (2022)] in our revised manuscript. We have modified the wrong citation in the latest revised manuscript.

Thanks for the suggestion. we have double-checked the English to polish the language throughout the manuscript. We have also carefully checked the whole manuscript to correct grammar/typo errors. Once all reviewer comments are solved, we will ask an English-language editing company for help.